# Visual-GRoup AFFEct Recognition (V-GRAFFER): A Unified Application for Real-Time Group Concentration Estimation in E-Lectures

**Andreas M. Triantafyllou * and George A. Tsihrintzis ***

Department of Informatics, University of Piraeus, 18534 Piraeus, Greece
* Correspondence: aandreasam@gmail.com (A.M.T.); gatsihrintzis@gmail.com (G.A.T.)

**Abstract:** This paper presents the most recent version of V-GRAFFER, a novel system that we have been developing for Visual GRoup AFFEct Recognition research. This version includes new algorithms and features, as well as a new application extension for using and evaluating the new features. Specifically, we present novel methods to collect facial samples from other e-lecture applications. We use screen captures of lectures, which we track and connect with samples during the duration of e-educational events. We also developed and evaluated three new algorithms for drawing conclusions on group concentration states. As V-GRAFFER required such complex functionalities to be combined together, many corresponding microservices have been developed. The current version of V-GRAFFER allows drawing real-time conclusions using the input samples collected from the use of any tutoring system, which in turn leads to real-time feedback and allows adjustment of the course material.

**Keywords:** visual group affect recognition; group affect recognition; emotion detection; group emotion detection; sentiment detection; group concentration; tutoring system



## 1. Introduction

In a series of recent research works [1–6], we have been presenting the progress of our development of a system called V-GRAFFER (standing for Visual GRoup AFFEct Recognition), which detects the emotional state of groups of people in real-time via processing visual data of theirs. More specific applications of our system include, among others, monitoring the emotional state of attendants of electronic lectures, Q&A sessions, and other educational events. The V-GRAFFER services and algorithms have been developed in an attempt to increase the quality of educational/tutoring systems and to detect and address difficulties that arise during each e-lesson based on particular group emotional states. V-GRAFFER can also help tutors to handle e-courses better. On the other hand, students are often faced with plenty of difficult/complex situations in their lives that affect them during their participation in e-educational events. Managing to automatically detect such situations, tutoring systems and human tutors become able to adapt each e-course material in real-time. In this way, the modified e-lecture/e-course flows can return better/optimal results regarding the overall learning of the students/participants.

Apart from e-course and lesson handling, drawing conclusions through group emotion detection can be used in many other fields in order to adapt corresponding content. For example, the emotion detection of a group of participants based on audience reactions can be used to evaluate performances, shows, concerts, music festivals, online meetings, recommendation-based smartphone applications, and other related events. Indeed, in a music event in which music tracks should be automatically selected, detecting group emotion will provide useful insight into the selection of the next music pieces and help adjust the playlist to be in harmony with the mood of the audience. This is particularly so as

audiences within the same event are usually very diverse, and real-time mood/sentiment detection can help track the audience's mood.

Emotion detection algorithms can combine several modalities, signals, and methodologies in order to achieve more accurate results. In the V-GRAFFER system, the group itself is the more significant part on which final conclusions are drawn. The entire group can affect each member individually, but each member can also affect the entire group. For this reason, in V-GRAFFER, we predict and investigate the group interaction based on members (i.e., participants who attend each event), time (i.e., specific time at each event), and depth of time (i.e., correlated moments one after another during the event, each specific time is associated with the previous and the following ones).

In this paper, we present three novel algorithms for calculating group concentration percentage at every moment, having set parameters about the sample intervals and time depth reference. Each algorithm is targeted at something different. This helps to evaluate them and decide which of these is more appropriate for drawing conclusions on groups of participants in e-educational events. The first algorithm is targeted on the average concentration values of each participant and next on the average of the entire team. In this algorithm, each random momentary lack of concentration has the same value as in other previous moments, regardless of when it happens. Thus, if a momentary lack of concentration arises, for example, 40 samples before the present, they are considered to burden exactly the same as if they arose in the present. In this way, a momentary lack of concentration at the present will have a limited influence on the total concentration value. However, in cases when the corresponding participant has begun to attend the event carefully only during the last moments, then the concentration value will stay reduced for some samples/time.

The second algorithm is adapted to use weight in each concentration value based on each time. Thus, the recent samples will have a higher value than the older ones. In the previous example, if the participant has begun to attend the e-course only during the last minutes, the previous lack of concentration will have a smaller effect on the concentration value. Finally, the third algorithm for group concentration value calculation uses a weighted average, as does the second algorithm, but the weighted average is applied twice, once on individual participants and once on the total group concentration values for a set interval. In this way, the set interval for the group concentration values ''holds'' and influences the values according to the group. All three of these algorithms will be analyzed in the following paper sections.

In the paper, we also present the software development process that incorporates the new features. Specifically, we developed services and application extensions in order to support the previously mentioned algorithms. Specifically, we implemented the new features which can track e-lectures/e-courses in an extension that is installed into our main Windows Presentation Foundation (WPF) application that we have developed regarding the V-GRAFFER system. Moreover, we have introduced the functionality for the three algorithms for group concentration value calculation. For this step of our work, we have created a set of app settings for samples and time intervals. These values are variable, and we can easily adjust them in order to adapt the application to different conditions and situations. We have tested with different settings in our experiments, and we present the results in Section 5.

The remainder of the paper is organized as follows: Section 2 summarizes previous related works in this research field. Section 3 analyzes our proposed algorithms for group concentration value calculation. The corresponding software implementations are described in Section 4. Section 5 contains experimental results from the use of the newly implemented features. Through these experiments, we aggregated results on each algorithm which we compare and evaluate in Section 6. Finally, Section 7 presents a paper synopsis and conclusions, as well as indications of future related work.

## 2. Previous Related Work

Much of the literature has appeared in the past two decades on the problem of automated detection of the emotional state of either individuals or groups of people from their visual, audio, or biological signals. With the early works by psychologists [7–11] as starting point, computer scientists have been showing strong interest in devising methodologies for the recognition of emotion in software users [12]. This research effort has been driven by the fact that emotion plays a very significant role in human-with-human interaction, which, in turn, may be exploited in human–machine interactions as well [13–17].

More specifically, early attempts at emotion recognition include the nominal works by Picard [18,19], De Silva et al. [12], and Pantic and Rothkrantz [17], who essentially defined the field of *affective computing*. A great portion of more recent research has been devoted to the detection of the emotional state of humans from visual-facial data, i.e., from images or videos of their faces, e.g. [20–22]. Some other works have addressed visual-facial emotion detection under special circumstances, such as when the subject suffers from depression [23] or is tired [1] or the detection of mild cognitive impairment [24]. Research has also been conducted on facial expressions according to stability over time [25] and related to the context for classification and form standardization [22,26]. Other approaches to the detection of emotion include alternative modalities, i.e., collection and processing of non-visual data. For example, successful approaches to emotion detection combine visual-facial modalities with audio-lingual modalities and modalities based on patterns in keyboard strokes [27]. Specifically, emotion detection based on keyboard stroke patterns has been investigated, among others, in [13], while linguistic and paralinguistic audio signals were considered, for example, in [28]. Similarly, meaningful information extraction from body posture or gait was considered; for example, in [29,30], emotional expression in gestures was investigated. For example, in [31], emotion detection from written texts and bio-signal measurements has also been investigated [32]. Today, the area of affective computing and automated emotion recognition in human–computer interaction applications remains an active research field, and the reader is pointed to recent survey works (for example, [33–35]) for further reading. In addition to research on emotion detection, related researches continue to be conducted on the very essence of human emotion and its intensity classification [36]. It is certain that the field of human emotion detection and corresponding affective computing application will continue to advance throughout the foreseeable future.

The detection of emotion in a group of people rather than in individuals is also very important [37], especially in events in which a number of individuals participate. In the recent past, researchers have looked into the group affect recognition problem [37,38] and corresponding developed systems [6,39]. In our previous related works [2–6], we have been developing a system that makes use of visual data (videos) of a group of participants and draws conclusions on the emotional state of the group. Specifically, we have been developing the V-GRAFFER system in which special emphasis has been placed on educational events and learning applications. Thus, our research so far has aimed at groups of students in various class settings. However, our research results and our system implementations may be used in several fields in which groups of people are involved.

In our most recent paper [6], the foundations of V-GRAFFER were described in detail, and its performance was illustrated and evaluated. Specifically, we (1) presented definitions, (2) analyzed our proposed approaches to useful data recognition, collection, and processing, (3) presented the database schemas in V-GRAFFER, and (4) described our experimental activities and software implementations and the corresponding evaluations.

In this paper, we present three novel algorithms for calculating group concentration percentages (Section 3). We also present the software development process that incorporates the new features and describes the (micro)services and application extensions that we have developed (Section 4). Moreover, we illustrate experiments performed with the current version of V-GRAFFER (Section 5) and evaluate its performance (Section 6). Finally, we summarize the paper, draw conclusions, and point to future research (Section 7).

## 3. Algorithms

The idea of group emotion detection requires a multifaceted algorithm approach in order to combine each participant of the group based on the entire team, time, depth of time, and the flow of each sample itself. For this reason, we devised three algorithms for calculating the group concentration values. These algorithms have some requirements in order to be usable under real conditions. These requirements include the sample detection and collection algorithms, the sample comparison and classification methods, the construction of training sets, and a core implementation for using them. We have implemented and presented some of these requirements in [1–6]. The remaining requirements will be presented in the following sections.

More specifically, in our previous works [1–6], we have implemented algorithms for automatic sample detection through pattern recognition, collecting them into flexible database schemes and classifying them via appropriate algorithms. We have presented processing algorithms for emotion detection using a comparison of expressions in samples of facial images with sample images already classified by experts. These samples are registered as vectors in the database and concern the related values of images of students who concentrate at each educational event. It is expected that a non-zero error rate will arise as this is an automated process using only facial samples. A point to emphasize is that by combining these algorithms with the current paper's work, we obtain results for students who no longer pay attention to the course or may have even left the (virtual) room. As V-GRAFFER is structured, essentially, any other facial expression detection algorithm can be used at this point. Furthermore, we have developed the main WPF application, which includes all these features, and we are now able to collect and create completed databases in order to use them as training sets at any relevant application. These functionalities are very important for our research work because training sets are to be used with group samples, along with the main algorithms for e-lecture/e-course tracking. As we present in the following paragraphs, we use these functionalities for creating training sets and for feeding the following algorithms. Thus, we now have available a unified application from which we can draw conclusions about group concentration percentage.

The first algorithm that we present is targeted at the calculation of the concentration values using balanced sample values during the time windows. In particular, we have set a time window, and the sample values which are included in that window have the same influence in each concentration value calculation. This means that an average of samples and time window returns the individual concentration values. For the group concentration value using the first algorithm, we calculate the average of the partial individual values.

For example,

**Scenario 1.** *We have four students who are attending an e-educational event, and one of them has missed the* **first** *30 s. The time window interval is set at 60 s, and we obtain one sample per second. We suppose that all students are attending the e-course with a concentration value of 100% except for the first student, who had a concentration value of 0% for the first 30 s and 100% for the rest. The settings for scenario 1 are presented in Table* 1*:*

**Table 1.** Scenario 1 settings for algorithms.

| Students | Time Window | Samples Per Second |
|:---:|:---:|:---:|
| 4 | 60 s | 1 |

1st student = 30 samples ∗ 0 (0% concentration value) + 30 samples ∗ 1 (100% concentration value)/60 (time window) = 30/60 = 0.5 ⇔ **50%**

So, the Group Concentration value will be:

3 students ∗ 100 (%) + 1 student ∗ 50 (%)/400 = 350/400 = 0.875 ⇔ **87.5%**

Consequently, if there was some momentary lack of concentration in some samples before the present, they will affect the current concentration values with the same value

weight. Thus, in cases of the participant beginning to attend the event carefully in the last moments, the concentration value will stay reduced for some samples/time.

The second algorithm that we present includes adding weight to recent and old samples. In this way, we manage to balance the time that a student lacks concentration placing more emphasis on the present samples. Thus, the calculation type is modified to:

$$\sum_{k}^{n} k a_n,$$

while the denominator becomes:

$$\sum_{k}^{n} k$$

Thus, the concentration value is calculated as follows:

$$P = \frac{\sum_{k}^{n} k a_n}{\sum_{k}^{n} k}$$

Using the second algorithm in the previously mentioned Scenario 1, we will have the following:

1st student = 1365/1830 = 0.745 ⇔ **74.5%**

Thus, the group concentration value will be:

3 students $*$ 100 (%) + 1 student $*$ 74.5 (%)/400 = 374.5/400 = 0.936 ⇔ **93.6%**

**Scenario 2.** *We have four students who are attending an e-educational event, and one of them has missed the **last** 30 s. We keep the setting of Scenario 1.*

We have:

1st student = 465/1830 = 0.254 ⇔ **25.4%**

The Group Concentration value will be:

3 students $*$ 100 (%) + 1 student $*$ 25.4 (%)/400 = 325.4/400 = 0.813 ⇔ **81.3%**

Thus, the recent samples will have a higher value than the older ones. Consequently, if the participant has begun to attend the e-course during only the last few minutes, the previous lack of concentration will have a lower effect on the concentration value.

The third algorithm for the group concentration values is a variation of the second weighted algorithm. The difference between the two algorithms lies in the fact that the third algorithm applies the second one twice: firstly, on the individual samples and, secondly, on the total group concentration using a second time window.

Applying this algorithm to Scenario 2 and keeping the previous settings, we need to set the group time/sample window. Let us set it to 5. The modified settings for third algorithm are presented in Table 2:

**Table 2.** Modified settings for third algorithm.

| Students | Time Window | Samples Per Second | Group Samples Window |
|----------|-------------|--------------------|----------------------|
| 4 | 60 s | 1 | 5 |

Furthermore, we suppose that the first 4 of the 5 group concentration values were 100%. Thus, we have:

1st student = 465/1830 = 0.254 ⇔ **25.4%**

The current group concentration value will be:

3 students $*$ 100 (%) + 1 student $*$ 25.4 (%)/400 = 325.4/400 = 0.813 ⇔ **81.3%**

The final group concentration value will be:

$((5-4) * 100 + (5-3) * 100 + (5-2) * 100 + (5-1) * 100 + (5-0) * 81.3)/1500$

= 1406.5/1500 = 0.937 ⇔ **93.7%**

Using this algorithm, values are "held" and influenced by the values of the group.

Each of these algorithms can be selected by the user in real time and the user may also switch between algorithms. In the next section, we analyze the software implementation in more detail.

## 4. Software Development and System Implementation

Along our research path, many software modules were developed in order to make group emotion detection appropriate for participants who attended e-educational events. Thus far, we have implemented many significant parts for each specific procedure. Specifically, we have presented in our previous works [1–6] software implementations for the automatic detection, collection, processing, and storage of group samples. Furthermore, we have implemented algorithms for automatic sample classification in order to keep them available during the educational event duration. Flexible database schemes have been designed and created, and evaluation algorithms have been developed and have accompanied our foundation system. Moreover, we have worked with some auxiliary implementations for testing and evaluation purposes. Most of these components have been developed into microservices in order for V-GRAFFER to be flexible and easy to amend and adapt.

During the technical part of the software implementation, we worked with many technologies in order to support the functionalities. Specifically, MS SQL Server and MySql have been used for database issues. Python and C# have been used for back-end development for our core and test services. Using MS Visual Studio, we have implemented our core service with WPF application technology. Finally, the APIs have been developed using the .Net core and .Net 6 frameworks. The latter can be released to both Linux and Windows Servers via the Docker tool, which allows us to have great flexibility.

In this paper, we present algorithms and user interface (UI) features that we have developed, supporting our devised and defined goals. First and foremost, the previously mentioned algorithms have been implemented in our services using local queues for concentration values for students and groups as buffer mechanisms. Furthermore, we have used functions and variables for adjusting parameters for algorithm intervals and limits.

For example, if we have set the limit of the sample to 60 and the sample interval to 1 sample per second, the buffer will appear as in Figure 1.

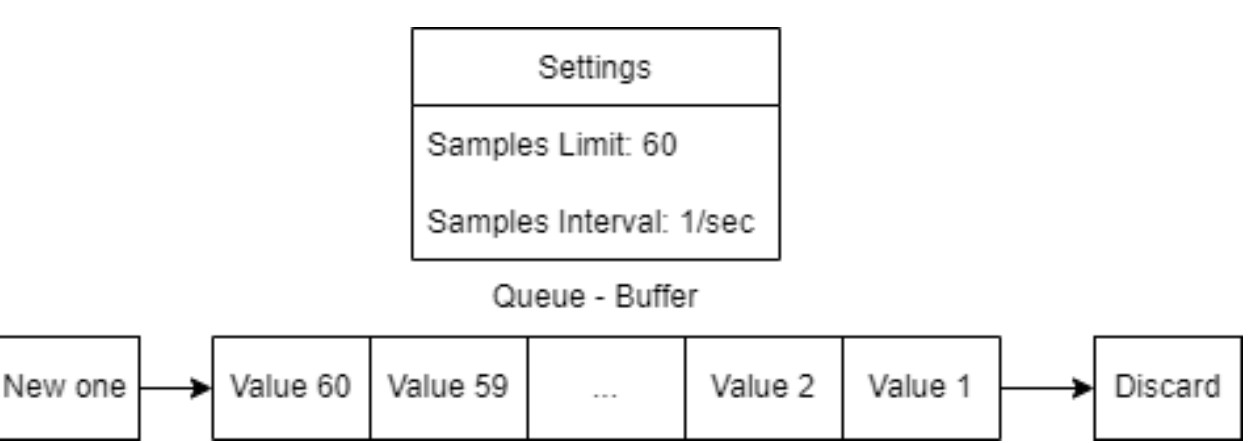

**Figure 1.** Queue–Buffer.

Moreover, we have implemented a feature for live sample capturing per each setting. This feature is useful for our software because our algorithms can be used with any visual e-learning/tutoring system.

Thus, the high-level system architecture chart has been adapted to Figure 2.

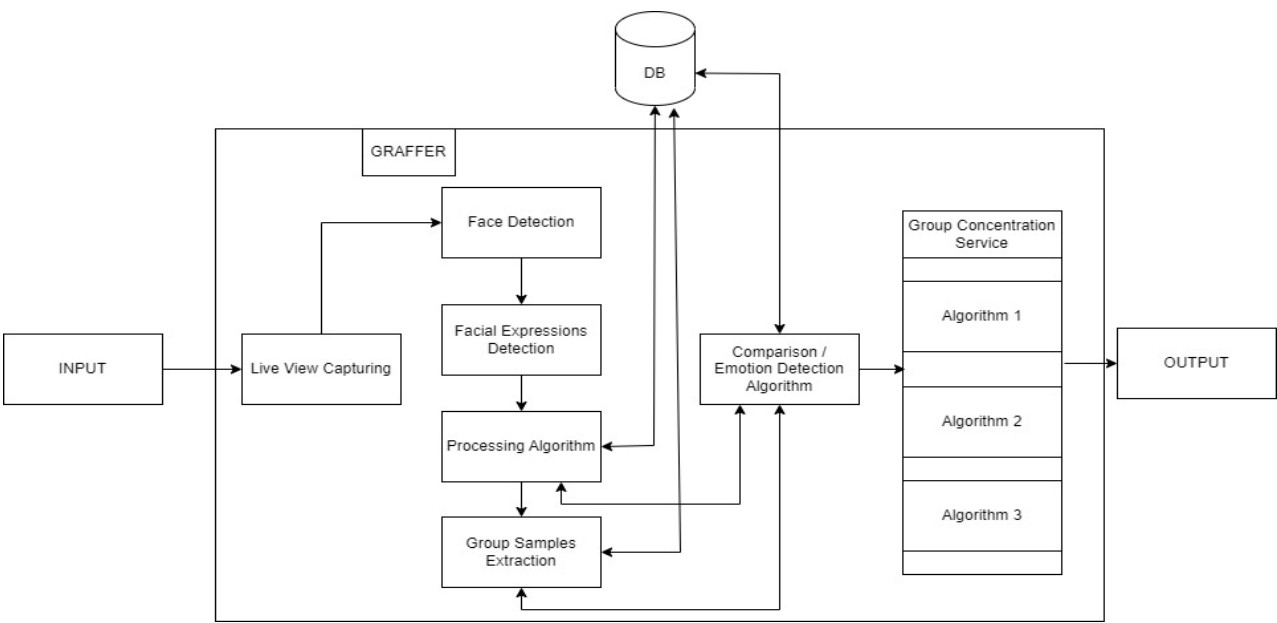

**Figure 2.** System architecture.

Apart from back-end algorithms and features, we needed UI changes in order to support our new functionalities. For this reason, we have added a new opinion for a new window named "Live Consultant" as in Figure 3:

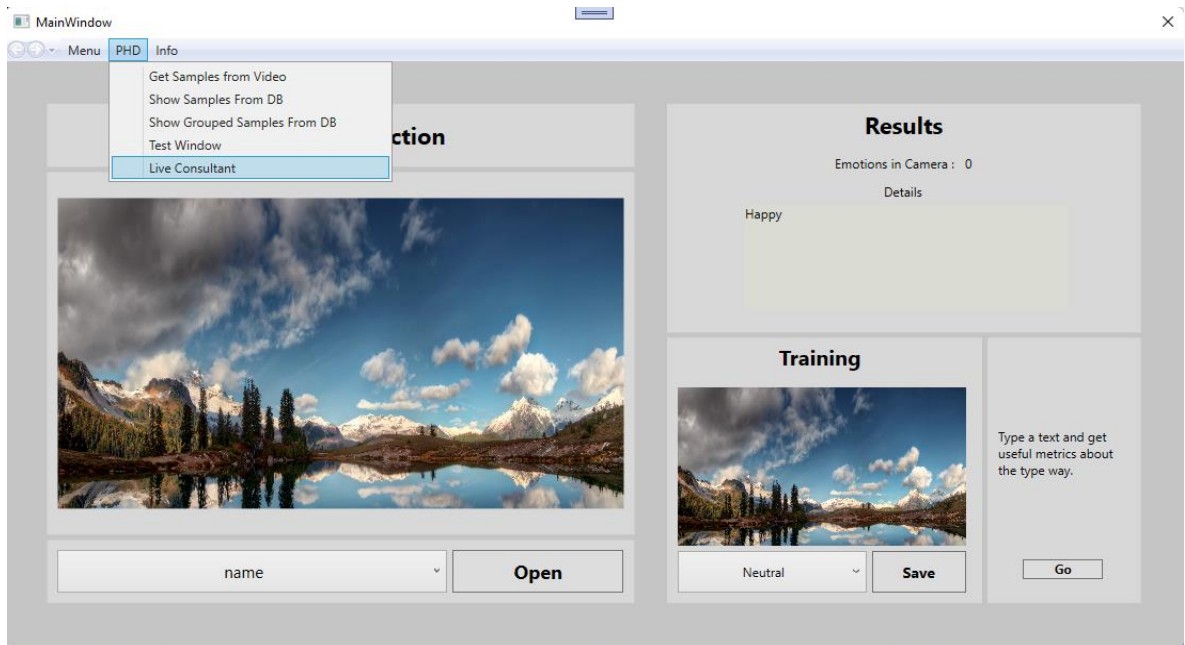

**Figure 3.** Live Consultant Option.

This new option opens a new window we have implemented for Live Consultants (Figure 4), in which we have new features for the group concentration algorithms. In particular, we have introduced a label for Live Concentration values with a color point for "high", "mid", and "low" values (green, orange, and red, respectively), a live line chart for values, one view for each event which happens, three radio buttons for the previously mentioned algorithms and buttons for functionality handling.

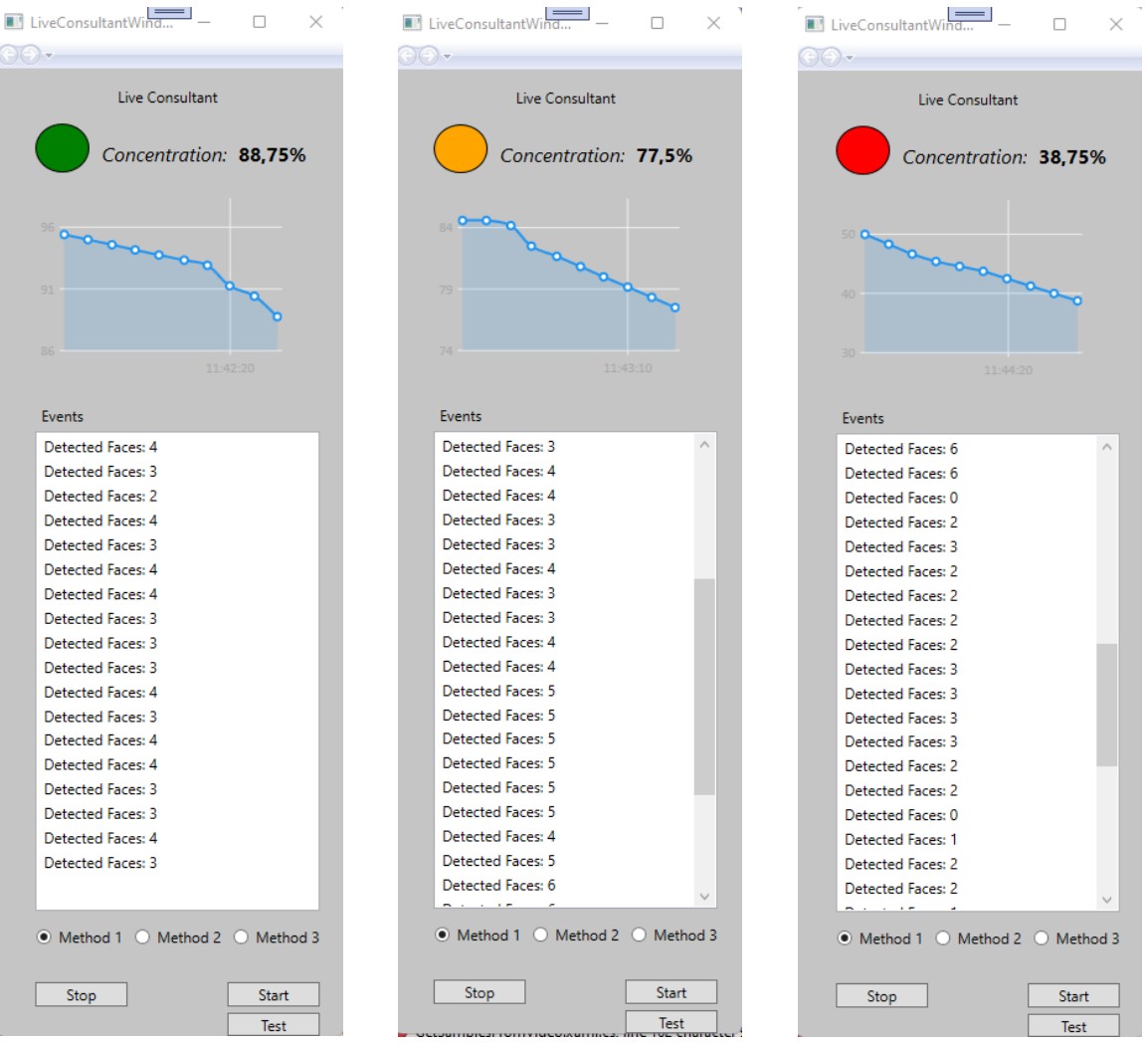

**Figure 4.** Live Consultant Window.

The new UI features are presented in Figure 5.

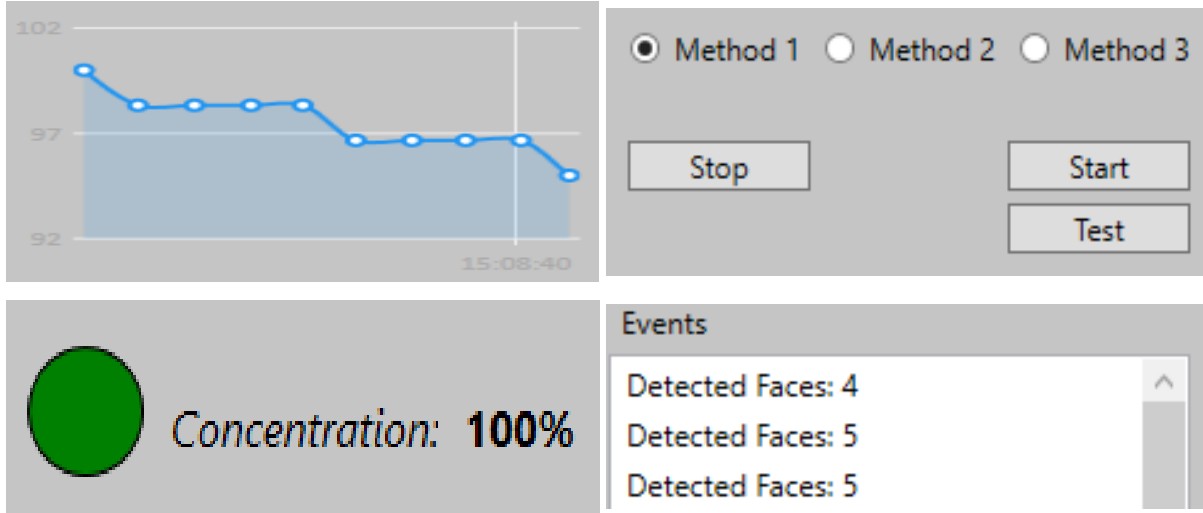

**Figure 5.** UI Features into Live Consultant Window.

Finally, we have created handler functions that update the UI controls according to each state. Through these software extensions and algorithms, we developed an integrated application that can detect and calculate the group concentration values via our algorithms from e-learning/e-course applications.

## 5. Experimental Activities

In this section of the paper, we present experimental activities we conducted in order to implement, test, and improve our algorithms for group concentration calculation. Specifically, our experimental activities consist of two parts. The first part includes recorded lectures that have been carried out at the University of Piraeus under real-life conditions. These recorded videos were used for sample collecting via our software managing tool to create a complete database of group samples. We selected some samples as a training set for our algorithms, and we commenced the test for improving processes. We repeated this procedure for various recorded lessons, and we managed to improve our algorithms by choosing various settings and thresholds for sample detection. A significant improvement came via correcting algorithm behavior in edge cases. The second part of our experimental activities includes targeted recordings that we made using remote communication applications and web cameras. We reproduced e-learning conditions, and we managed to adapt each video to different student concentration conditions in different lesson parts.

All these samples were saved into an MS SQL Server for all different facial expressions, as in Figure 6. As we mentioned in our previous works [5,6], we have used flexible database schemas, which can be used with various algorithms and modules. Furthermore, the classification algorithms have grouped our samples, and training sets have been created with samples correlated in time, depth of time, and educational event, as in Figure 7.

In relation to the second part of our experimental activities, we have chosen the same Scenario 1 defined earlier for tests and evaluation purposes.

**Scenario 1 (previously defined).** *We assume that four students attend an e-educational event (Figure 8) for about 5 min. We have separated the scenario into some time parts. The students are attending the e-educational event according to Table 3:*

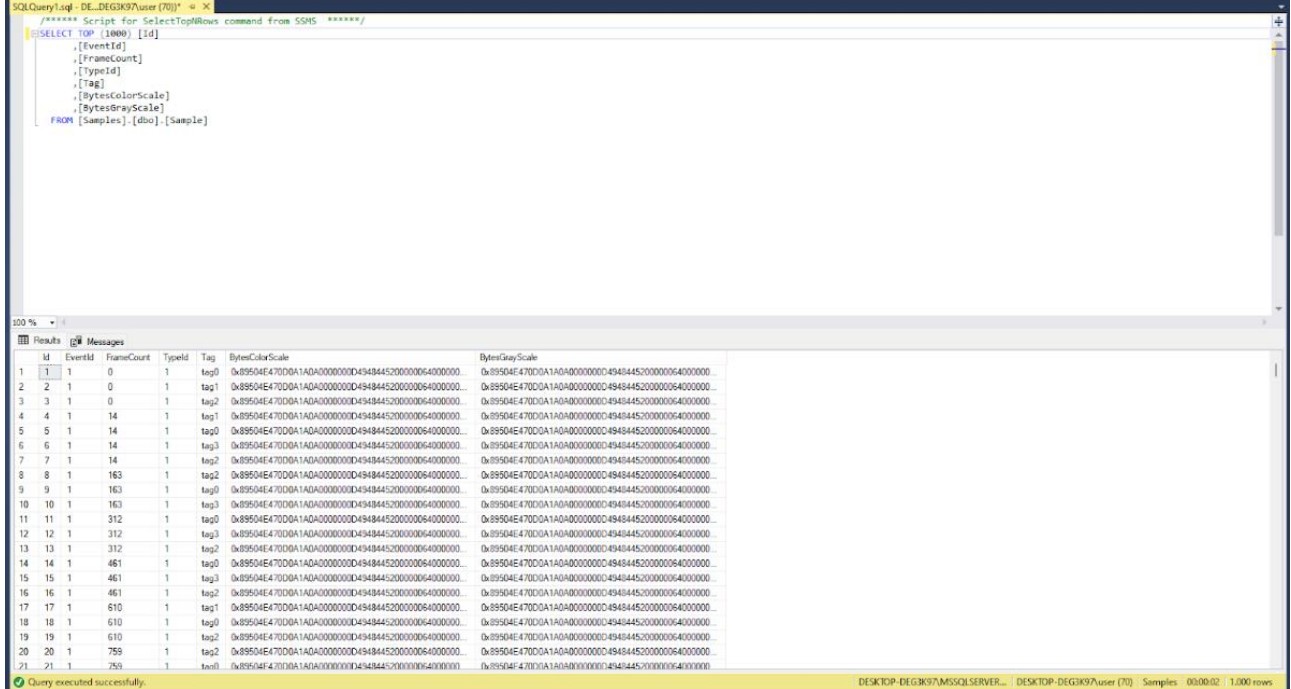

**Figure 6.** SQL Query for 1000 records–samples into SQL Server.

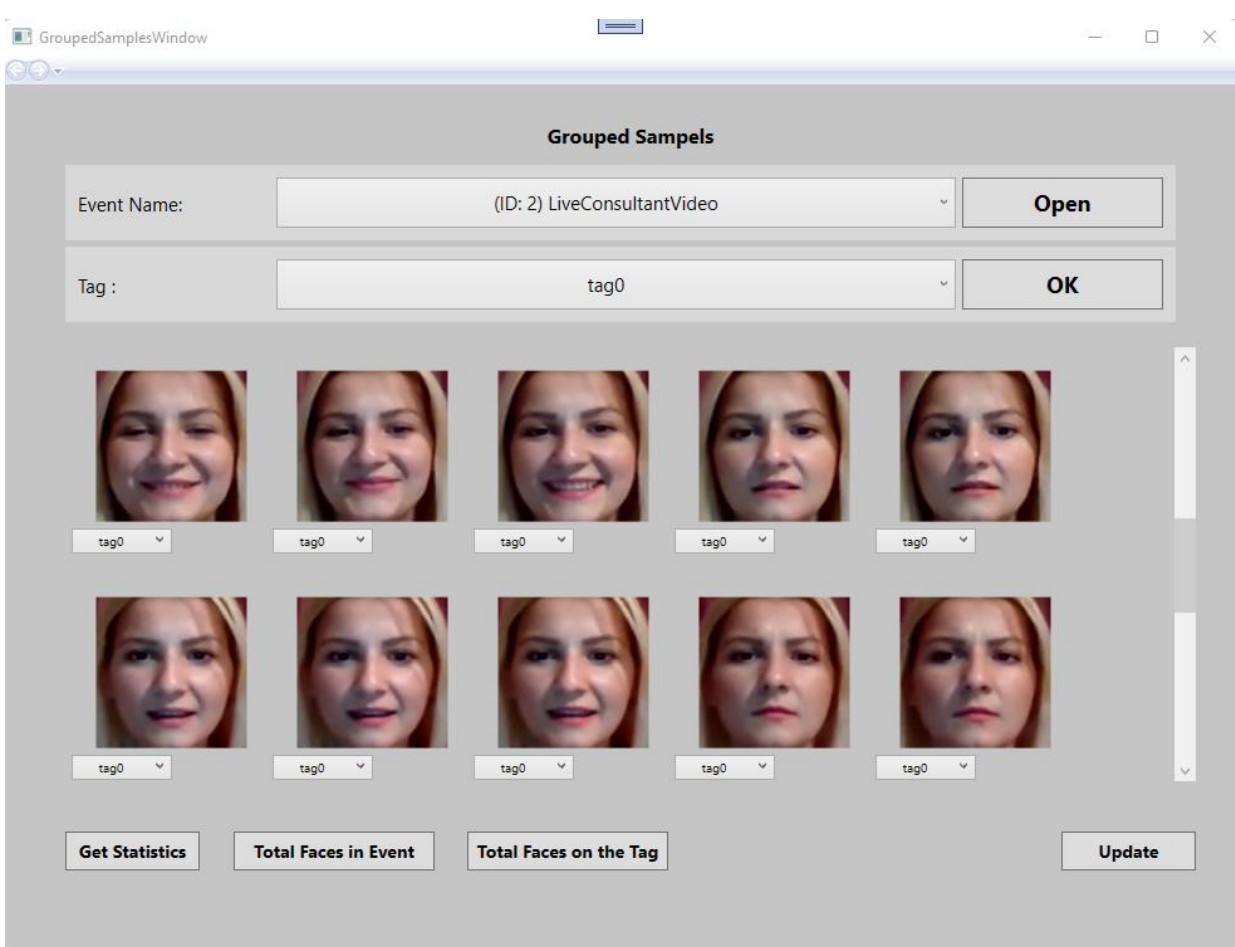

**Figure 7.** Grouped Samples.

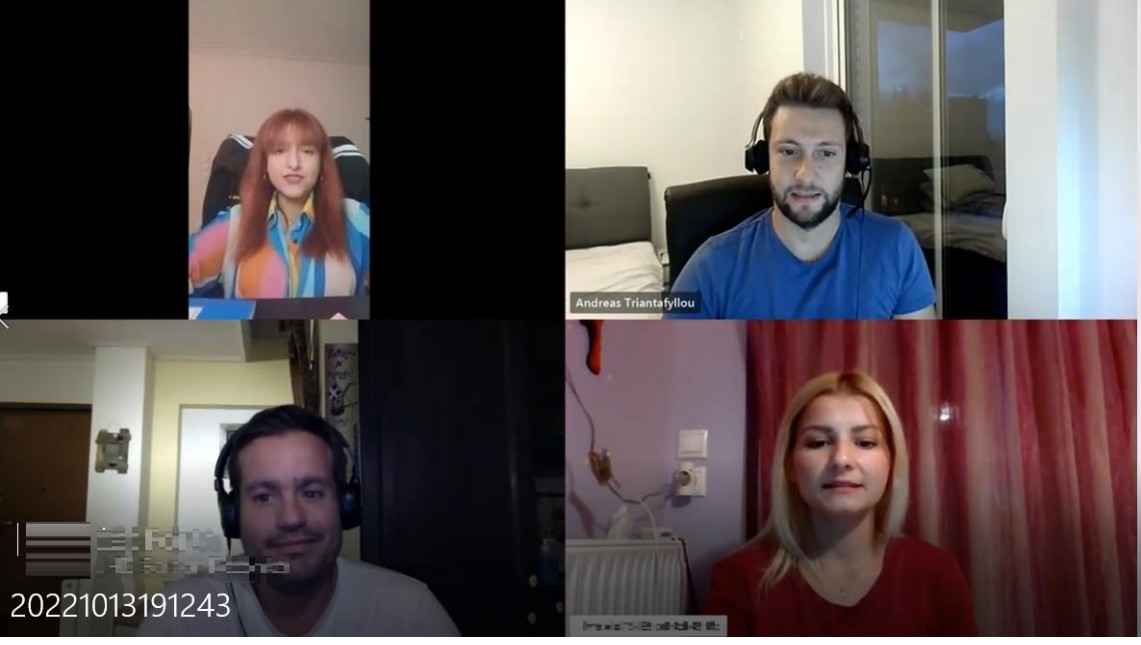

**Figure 8.** Four people into adjusted e-educational event.

**Table 3.** Time slots for student attendance.

| States | Time/Student | Student 1 | Student 2 | Student 3 | Student 4 |
|--------|-------------|-----------|-----------|-----------|-----------|
| **State 1** | ~40 s | ☑ | ☑ | ☑ | ☒ |
| **State 2** | ~40 s | ☑ | ☑ | ☑ | ☑ |
| **State 3** | ~30 s | ☒ | ☒ | ☑ | ☑ |
| **State 4** | ~30 s | ☑ | ☒ | ☒ | ☒ |
| **State 5** | ~1 min | ☒ | ☒ | ☒ | ☒ |
| **State 6** | ~1 min | ☑ | ☑ | ☑ | ☑ |

☒ = Student does not attend; ☑ = Student attends.

**Note:** We have set states 5 and 6 to have similar times as we wanted to reproduce and present the algorithm results after a longer processing circle of the selected threshold values. Specifically, we observed that, for state 5, the algorithm results were 0% or very close to 0%, as all four participants were not attending the course for an entire minute. On the other hand, all four participants attended the course at the entire last minute, which led to a concentration value very close to 100% (as we can see in the following sections).

Having recorded the video for Scenario 1, we used it for training set creation using V-GRAFFER. Thus, we selected the training set, and we proceeded with the group concentration value analysis. Examples of the samples of the selected training set are presented in Figure 9.

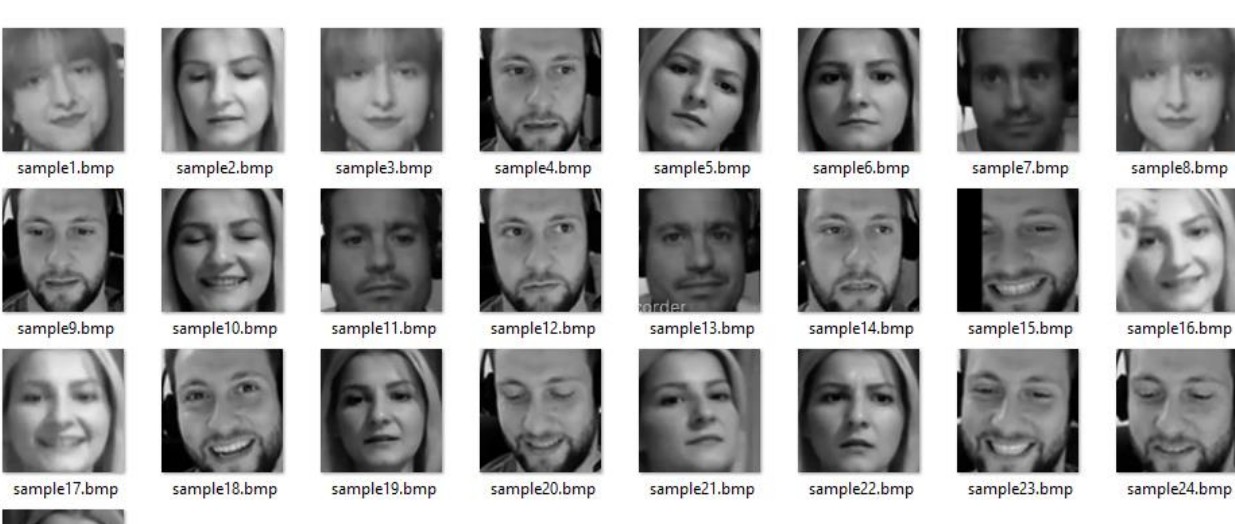

**Figure 9.** Selected Training Set.

## 6. Evaluations

After the completion of experimental activities, we proceed to the evaluation section. In particular, we used the recorded video for Scenario 1, and we evaluated the three new algorithms for group concentration value calculation.

The three algorithms have been run for the entire duration of the video, with each measurement taken at the same point in time for all algorithms. Here we can see the group concentration values per algorithm for all states of Table 3.

As we can observe in Table 4 and Figure 10, the second algorithm with time weights exhibits sharper shifts for group concentration values because the time when a student is or is not paying attention to the lesson carries more weight than the random times with a lack of concentration. Furthermore, we notice that, when using the third algorithm, the con-

centration values change more intensely at the moments that each Group Window is full, and the values "are held" by the group for some samples/period (Group Time Window) (Figure 11) as we mentioned earlier in the algorithms section.

**Table 4.** Group Concentration percentage per time slot.

| Algorithm/States | State 1 | State 2 | State 3 | State 4 | State 5 | State 6 |
|---|---|---|---|---|---|---|
| Alg. 1 | 75% | 81.25% | 67.08% | 34.58% | 0% | 99.58% |
| Alg. 2 | 68.56% | 86.13% | 55.46% | 28.12% | 0% | 100% |
| Alg. 3 | 74.44% | 82.77% | 60.51% | 30.42% | 0.10% | 98.37% |

**Figure 10.** Line Chart for Group Concentration values per Algorithm.

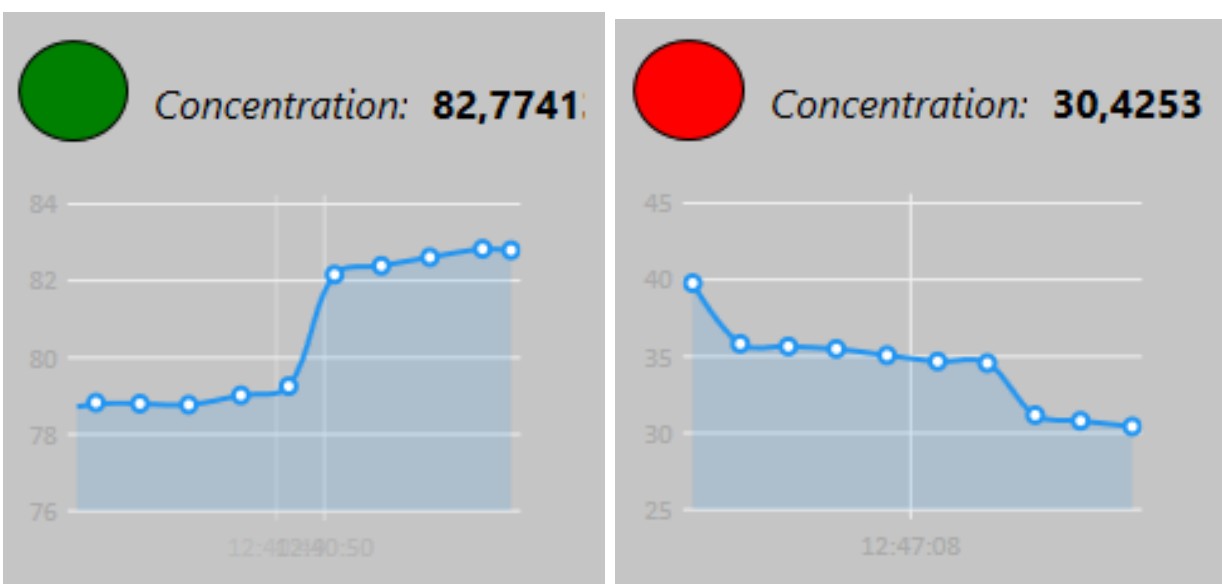

**Figure 11.** Charts for algorithm 3. Large steps per Group Window.

We can adapt each setting in order to make the Group Concentration calculation faster using different sample intervals and windows. This feature is very important because we can adapt and use it in different kinds of e-educational events. For example, we need a

faster calculation of group concentration for short sessions like a Q&A session because the students should be focused on the e-lesson for less time.

Finally, we propose the second algorithm, which uses time weights as a general solution, but the other two algorithms should be more suitable for specialized cases.

## 7. Summary, Conclusions, and Future Work

In summary, in this paper, we presented the new and integrated version of V-GRAFFER, our system for Visual GRoup AFFEct Recognition. This improved version allows us to calculate group concentration values in real-time using any video communication application. We connected our previous works, including the core application as well as the microservices for information extraction, sample recognition, and collection and classification, with the new functionalities, including UI features and the group concentration calculation algorithms. Specifically, we presented three algorithms with balanced, weighted, and group-influenced points regarding group concentration calculation. We evaluated them in order to reach a conclusion about the best response in real e-learning/e-course situations. Through evaluation, we came up with the proposal to use the second algorithm as a solution in general situations in various e-educational events. The other two algorithms can be used in specialized cases on the grounds that we may need balanced or group-influenced concentration calculations. Moreover, we mentioned the sample period and interval settings which can be adapted for different calculation speeds depending on distinct types of educational events.

The core of our research to date has managed to create services for improving the quality of e-educational events via the detection of the emotional/sentimental states of each group of participants in e-educational events. Another major achievement is the integration of V-GRAFFER services with e-tutoring systems in order to adapt lesson content and materials according to group sentiment. Furthermore, our developed services can be used as utilities for tutors in e-learning systems.

A future direction of our research includes the extension of the current version of V-GRAFFER in order to support more modules in addition to facial sample analysis. Moreover, we will enhance the facial feature extraction, thus improving the algorithms that we have implemented so far. We will investigate further aspects of group emotion detection by comparing different modules in different situations. Evaluations and conclusions will be drawn for combined and separate modules in order to decide the ideal conditions for each combination with regard to distinct e-educational events. Aspects of this research, as well as other related issues, are currently in progress, and its results will be presented elsewhere in the near future.

**Author Contributions:** Conceptualization, A.M.T. and G.A.T.; methodology, A.M.T.; software, A.M.T.; validation, A.M.T. and G.A.T.; formal analysis, A.M.T. and G.A.T.; investigation, A.M.T. and G.A.T.; resources, A.M.T.; data curation, A.M.T.; writing—original draft preparation, A.M.T. and G.A.T.; writing—review and editing, A.M.T. and G.A.T.; visualization, A.M.T.; supervision, G.A.T.; project administration, A.M.T. and G.A.T.; funding acquisition, G.A.T. All authors have read and agreed to the published version of the manuscript.

**Funding:** This research received no external funding.

**Institutional Review Board Statement:** No ethical approval is required.

**Informed Consent Statement:** Written informed consent has been obtained from the patient(s) to publish this paper.

**Data Availability Statement:** Not applicable.

**Conflicts of Interest:** The authors declare no conflict of interest.

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
