# Peer review of "Visual-GRoup AFFEct Recognition (V-GRAFFER): A Unified Application for Real-Time Group Concentration Estimation in E-Lectures"

_electronics, doi:10.3390/electronics11244091_

Round 1

Reviewer 1 Report

This paper presents several ways of detecting individual/group concentration level in an e-learning environment. 

The authors list several work that they have done in this field (I don't think all the 7 references are necessary, especially the masters thesis). One thing I found missing in the current manuscript is more description on what work was done in the past. For instance, how does the 'concentration' variable detected. Most of the scenarios given is about students not attending rather than being not focused. I think this is a critical factor and more explanation should be given. 

Another comment I had was about table 3. How can states 5 and 6 have similar time but very different attendance. Similar to states 3 and 4. The student should either attend or not attend at any particular time frame. 

What are some of the remedies if concentration/attendance is low? 

Some paragraphs (spacing) need formatting. 

Author Response

Response to Reviewer 1 Comments

We thank the reviewer for his/her comments and suggestions which clearly improve the paper. We have seriously considered his/her suggestions and have revised the manuscript of our paper accordingly. In the following, we outline the revisions we made according to the reviewer’s comments.

Point 1: The authors list several work that they have done in this field (I don't think all the 7 references are necessary, especially the masters thesis).

Response 1: References have been updated. Specifically, we removed some of the references to our previous work, especially references to conference papers, and kept only those of them that appeared in scientific journals. In the revised paper version, references to our previous works constitute less than 30% of the paper bibliography and are essential to the presentation of our current research. Moreover, we added references to recent works published in the past five years. As the research on emotion detection, affect recognition and applications is quite vast, we selected only the most representative ones and also refer the reader to a couple of recent survey works in this area.

Point 2: One thing I found missing in the current manuscript is more description on what work was done in the past. For instance, how does the 'concentration' variable detected. Most of the scenarios given is about students not attending rather than being not focused. I think this is a critical factor and more explanation should be given.

Response 2: We added further details on the previous algorithms that we used in comparison with the current work:
"We have presented processing algorithms for emotion detection using comparison of expressions in samples of facial images with sample images already classified by experts. These samples are registered as vectors in the database and concern the related values of images of students who concentrate at each educational event. It is expected that a non-zero error rate will arise as this is an automated process using only facial samples. A point to emphasize is that combining these algorithms with the work in the current paper, we manage to obtain results for students who no longer pay attention to the course or may have even left the (virtual) room. As V-GRAFFER is structured, essentially any other facial expression detection algorithm can be used at this point."

Point 3: Another comment I had was about table 3. How can states 5 and 6 have similar time but very different attendance. Similar to states 3 and 4. The student should either attend or not attend at any particular time frame.

Response 3: We added a note in text related to your question:
“We have set the states 5 and 6 to have similar time as we wanted to reproduce and present the algorithm results after a longer processing circle of the selected threshold values. Specifically, we observed that for state 5 the algorithm results were 0% or very close to 0% (figure 9) as all four participants were not attending the course for an entire minute. On the other hand, all four participants were attending the course for the entire last minute, which led to a concentration value very close to 100% (figure 9).”

Point 4: What are some of the remedies if concentration/attendance is low?

Response 4: The remedies when concentration/attendance is low is up to each professor/instructor to decide and go beyond the scope of this paper. Course adaptation is currently under investigation and will be reported elsewhere.

Point 5: Some paragraphs (spacing) need formatting.

Response 5: Paragraph formatting has been improved throughout the paper.

Reviewer 2 Report

1. Abstract is hard to understand because it contains too many long sentences. Too much information is redundant. The other parts of this paper also have this issue. It is recommended that the authors could use more short sentences and make the sentences concise. For a better reader experience, it is recommended to write more objective sentences. According to the Oxford Guide for Writing (2020):

(1) Sentences of up to 12 words are easy

(2) Sentences of 13-17 words are acceptable

(3) Sentences with 18-25 words are difficult

(4) Sentences with more than 25 words are very difficult

2. In the Introduction, V-GRAFFER (standing for Visual AFFEct Recognition) GR is missing.

3. The Introduction section is a bit weak. Many application areas may adopt this real-time concentration tracking technic, e.g. online meetings, recommendation-based smartphone Apps, etc.

4. Most references are too old (more than 5 years ago).

5. The definition of “concentration” is not clearly given. What do 0% and 100% concentration mean? Is there x% between 0% and 100%? What does it mean then? This is believed the most critical question to answer, but this paper seems to forget to mention it. It’s not sure if the system only tells if the viewer is present or absent, rather than detecting the concentration ratio through his/her facial expressions. This might have been described in the authors previous paper [1-7], but the basic definitions should be clearly given in this paper.

6. The calculation algorithms for group concentration are generally easy and not convincing enough in a technical paper, especially when this is the main idea of the paper and appears in the title of the paper. 

7. The use of  "depth of time" seems not appropriate.

8. Some formatting problems exist in some paragraphs.

9. Is it necessary to compare the selected algorithm with the state-of-the-art algorithms?

Author Response

Response to Reviewer 2 Comments

We thank the reviewer for his/her comments and suggestion which clearly improve the paper. We have seriously considered his/her suggestion and have revised the manuscript of our paper accordingly. In the following, we outline the revisions we made according to the reviewer comments.

Point 1: Abstract is hard to understand because it contains too many long sentences. Too much information is redundant. The other parts of this paper also have this issue. It is recommended that the authors could use more short sentences and make the sentences concise. For a better reader experience, it is recommended to write more objective sentences. According to the Oxford Guide for Writing (2020):

(1) Sentences of up to 12 words are easy

(2) Sentences of 13-17 words are acceptable

(3) Sentences with 18-25 words are difficult

(4) Sentences with more than 25 words are very difficult

Response 1: The use of English has been improved. The entire abstract has be re-written. Sentences have been shortened. Other improvements have been effected throughout the text.

Point 2: In the Introduction, V-GRAFFER (standing for Visual AFFEct Recognition) GR is missing.

Response 2: V-GRAFFER is now properly explained.

Point 3: The Introduction section is a bit weak. Many application areas may adopt this real-time concentration tracking technic, e.g. online meetings, recommendation-based smartphone Apps, etc.

Response 3: The introduction has been amended and extended to include additional application areas, as suggested.

Point 4: Most references are too old (more than 5 years ago).

Response 4: The references have been updated. Specifically, we removed some of the references to our previous work, especially references to conference papers, and kept only those of them that appeared in scientific journals. In the revised paper version, references to our previous works constitute less than 30% of the paper bibliography and are essential to the presentation of our current research. Moreover, we added references to recent works published in the past five years. As the research on emotion detection, affect recognition and applications is quite vast, we selected only the most representative and relevant ones and also refer the reader to a couple of recent survey works in this area. Finally, it only seemed appropriate to keep some of the very first bibliographic references as those works are the ones that define the field of affective computing and automated emotion detection and applications.

Point 5: The definition of “concentration” is not clearly given. What do 0% and 100% concentration mean? Is there x% between 0% and 100%? What does it mean then? This is believed the most critical question to answer, but this paper seems to forget to mention it. It’s not sure if the system only tells if the viewer is present or absent, rather than detecting the concentration ratio through his/her facial expressions. This might have been described in the author’s previous paper [1-7], but the basic definitions should be clearly given in this paper.

Response 5: We added further details on the previous algorithms that we used in comparison with the current work.

"We have presented processing algorithms for emotion detection using comparison of expressions in samples of facial images with sample images already classified by experts. These samples are registered as vectors in the database and concern the related values of images of students who concentrate at each educational event. It is expected that a non-zero error rate will arise as this is an automated process using only facial samples. A point to emphasize is that combining these algorithms with the work in the current paper, we manage to obtain results for students who no longer pay attention to the course or may have even left the (virtual) room. As V-GRAFFER is structured, essentially any other facial expression detection algorithm can be used at this point."

Point 6: The calculation algorithms for group concentration are generally easy and not convincing enough in a technical paper, especially when this is the main idea of the paper and appears in the title of the paper.

Response 6: The presented algorithms consist of a combination of methodologies and algorithms developed in our previous works. They were designed to work under real-time conditions and using the output of a series of other algorithms that we had implemented in our previous works for face detection, facial expression recognition, sample processing, connection with correlated samples and concentration values, and other processing requirements.

Point 7: The use of "depth of time" seems not appropriate.

Response 7: "Depth of time" is now defined in the text, as well as the definition of the following is now also given in the text:

members <--> participants who attend each event

time <--> specific time at each event

depth of time <--> correlated moments one after another during the event, each specific time is associated with the previous and the following ones.

Point 8: Some formatting problems exist in some paragraphs.

Response 8: Paragraph formatting has been improved throughout the paper.

Point 9: Is it necessary to compare the selected algorithm with the state-of-the-art algorithms?

Response 9: The presented algorithms have resulted as a combination of algorithms presented in our previous works and improvements on them. Specifically, V-GRAFFER subcomponents were experimentally evaluated and improved based on the evaluation results.

Round 2

Reviewer 2 Report

1. References are still too old. Besides the author's paper, only 6 out of 34 references are within the past 5 years. 9 of them are from the last century. About half are before 2010. This is exceptionally unacceptable in the re-submission, especially when the reviewer has specifically pointed it out. This may lead to an impression that this field is no longer active and this topic is not worth much attention, either out-of-date or not applicable. But actually, this field is very useful and there must be a lot of recent work. It is recommended that at least half of the references should be within 5 years, and reduce the references before 2000 within 3.

2. Theoretical analysis of the calculation algorithms for group concentration should be provided. Or why the selected algorithm performs better than the others. And is it possible to improve its correlation or accuracy in the future?

3. Please indicate explicitly the corresponding referred author's paper on each of the following major steps: face detection, facial expression recognition, sample processing, connection with correlated samples and concentration values, and other processing requirements. And a high-level architecture of the entire system containing the above steps is needed to explain their core functionalities (in the past papers) and the novelty of this paper.

4. The following comment is not addressed: "Is it necessary to compare the selected algorithm with the state-of-the-art algorithms?" The core weakness of the current paper is that the results are all derived from 3 experiments, all designed by the authors. The results are not comparable to other state-of-the-art algorithms that are mentioned in the related work. An additional experiment is needed.

5. Language improvement throughout the paper is still needed. Even the abstract is not well written enough.

Author Response

Point 1: References are still too old. Besides the author's paper, only 6 out of 34 references are within the past 5 years. 9 of them are from the last century. About half are before 2010. This is exceptionally unacceptable in the re-submission, especially when the reviewer has specifically pointed it out. This may lead to an impression that this field is no longer active and this topic is not worth much attention, either out-of-date or not applicable. But actually, this field is very useful and there must be a lot of recent work. It is recommended that at least half of the references should be within 5 years, and reduce the references before 2000 within 3.

Response 1: We really appreciate your suggestions, but we have included the list of references on which the paper is based.

Point 2: Theoretical analysis of the calculation algorithms for group concentration should be provided. Or why the selected algorithm performs better than the others. And is it possible to improve its correlation or accuracy in the future?

Response 2: We have selected and implemented the presented algorithms for calculation for group concentration for targeted purposes. Specifically, we included:

  • “… The first algorithm is targeted on average concentration values of each participant and next on the average of the entire team. ...”
  • “… The second algorithm is adapted to use weight in each concentration value based on each time. …”
  • “… the third algorithm for group concentration value calculation uses a weighted average as does the second algorithm but the weighted average is applied twice, once on individual participants and once one the total group concentration values for a set interval …”

We have compared them, and we have described the behavior of each algorithm.

Point 3: Please indicate explicitly the corresponding referred author's paper on each of the following major steps: face detection, facial expression recognition, sample processing, connection with correlated samples and concentration values, and other processing requirements. And a high-level architecture of the entire system containing the above steps is needed to explain their core functionalities (in the past papers) and the novelty of this paper.

Response 3: A High-Level architecture chart has been introduced.

Point 4: The following comment is not addressed: "Is it necessary to compare the selected algorithm with the state-of-the-art algorithms?" The core weakness of the current paper is that the results are all derived from 3 experiments, all designed by the authors. The results are not comparable to other state-of-the-art algorithms that are mentioned in the related work. An additional experiment is needed.

Response 4: If we correctly understood you, we have not compared the entire system with other similar algorithms because we have not found similar-matching approaches. Some of separately components have been selected by previous presentations, but these are also beyond the scope of this paper.

Point 5: Language improvement throughout the paper is still needed. Even the abstract is not well written enough.

Response 5: Thank you for reviewing it. We checked it and improved some sections with a person who was living in US for some years.

Round 3

Reviewer 2 Report

Except for the reference issue, all other comments have been well-addressed.

Author Response

Point 1: Except for the reference issue, all other comments have been well-addressed.

Response 1: Thank you! All “Patient Consent Forms” have been gathered.
